# Natural Products as the Potential to Improve Alzheimer’s and Parkinson’s Disease

**DOI:** 10.3390/ijms24108827

**Published:** 2023-05-16

**Authors:** Sung Wook Kim, Jun Ho Lee, Bumjung Kim, Gabsik Yang, Jong Uk Kim

**Affiliations:** 1College of Korea Medicine, Woosuk University, Jeonju-si 54986, Republic of Korea; sheep.sw91@gmail.com (S.W.K.); celtece@daum.net (J.H.L.); 2Da Capo Co., Ltd., Jeonju-si 54986, Republic of Korea; 3Department of Oriental Health Management, Kyung Hee Cyber University, Seoul 02447, Republic of Korea; oripharm@khcu.ac.kr

**Keywords:** Alzheimer’s disease, Parkinson’s disease, neurodegeneration, natural products

## Abstract

Alzheimer’s disease and Parkinson’s disease are the two most common neurodegenerative diseases in the world, and their incidence rates are increasing as our society ages. This creates a significant social and economic burden. Although the exact cause and treatment methods for these diseases are not yet known, research suggests that Alzheimer’s disease is caused by amyloid precursor protein, while α-synuclein acts as a causative agent in Parkinson’s disease. The accumulation of abnormal proteins such as these can lead to symptoms such as loss of protein homeostasis, mitochondrial dysfunction, and neuroinflammation, which ultimately result in the death of nerve cells and the progression of neurodegenerative diseases. The medications currently available for these diseases only delay their progression and have many adverse effects, which has led to increased interest in developing natural products with fewer adverse effects. In this study, we selected specific keywords and thesis content to investigate natural products that are effective in treating Alzheimer’s and Parkinson’s diseases. We reviewed 16 papers on natural products and found that they showed promising mechanisms of action such as antioxidant, anti-inflammatory, and mitochondrial function improvement. Other natural products with similar properties could also be considered potential treatments for neurodegenerative diseases, and they can be consumed as part of a healthy diet rather than as medicine.

## 1. Introduction

Neurodegenerative diseases occur when nerve cells in the brain and spinal cord deteriorate, leading to abnormal functioning and eventual cell death. This can be triggered by various types of stress and inflammatory responses. Examples of neurodegenerative diseases include Alzheimer’s disease, Parkinson’s disease, amyotrophic lateral sclerosis, and Huntington’s disease, which are often caused by apoptosis [1]. Alzheimer’s disease is the most common neurodegenerative disease globally, affecting around 24 million people worldwide, with a sharply increasing incidence in those over 65 years of age [2]. Frontotemporal dementia, on the other hand, occurs mainly in people under 65 years of age, with an average onset age of 56 years and an incidence rate of approximately 17 cases per 100,000 people [3]. Parkinson’s disease is the second most common neurodegenerative disease after Alzheimer’s disease, with an incidence rate of 1–2 per 1000 people and affecting 1% of the population over 60 years of age [4]. Multiple sclerosis is the most common demyelinating disease in high-income countries, with a prevalence of over 100 per 100,000 in North America and Europe, compared to 2 per 100,000 in East Asia and Africa [5]. Huntington’s disease has varying prevalence rates globally, ranging from about 6 per 100,000 in North America to 0.42 per 100,000 in Asian countries [6]. Given the significant social and economic burden of Alzheimer’s and Parkinson’s diseases, which affect many patients, we will focus our investigation on these two diseases.

As research progresses, the exact cause of Alzheimer’s and Parkinson’s diseases remains unknown. However, there is a growing body of evidence that suggests that the disease is caused by a complex interplay of environmental and genetic factors [7,8]. Alzheimer’s disease is thought to be caused by neuronal cell death due to the formation of β-amyloid plaques or neurofibrillary tangles caused by the hyperphosphorylation of tau in neurons in the hippocampus of the brain. This is considered to be the primary cause [9]. On the other hand, Parkinson’s disease is associated with Lewy bodies formed by abnormal aggregation of α-synuclein in neurons in the substantia nigra of the brain, resulting in neuronal cell death [10]. In addition to the production of incorrect proteins, various causes such as neuroinflammation caused by external stress, oxidative damage, and mitochondrial dysfunction have also been reported [11]. Alzheimer’s disease is characterized by neurological symptoms such as decreased memory, slurred speech, and cognitive function [7]. Meanwhile, Parkinson’s disease is associated with non-motor neurological abnormalities such as olfactory dysfunction, cognitive impairment, psychosis, and sleep disturbance, as well as motor abnormalities such as tremor, rigidity, and gait disturbance due to abnormalities in the autonomic nervous system [8].

Despite ongoing research into the causes and treatments of Alzheimer’s and Parkinson’s diseases, there is currently no medication available that can fully cure them. While there are some medications that can alleviate or slow down the progression of these diseases by modulating certain neural mechanisms, they often have unwanted adverse effects as they affect other signal transduction pathways [12]. Therefore, there is a need to develop new medications that can overcome these limitations and adverse effects. To achieve this, researchers are turning to natural resources to find new and effective medications with improved safety profiles. In this study, we aim to investigate the potential of plants and plant-derived extracts for improving neurodegenerative diseases.

## 2. Neurodegenerative Diseases

Neurodegenerative diseases result from a complex interplay between environmental and genetic factors. Protein abnormalities are a significant factor in both diseases, and the ability to regulate protein function is critical. As individuals age, protein homeostasis gradually deteriorates, and the prevalence of neurodegenerative diseases increases. In addition to age-related environmental factors, genetic mutations can also disrupt normal protein production. Molecular changes observed in patients with neurodegenerative diseases include protein homeostasis loss, mitochondrial dysfunction, cellular aging, oxidative stress, and inflammation, which contribute to the disease’s pathology [13]. The objective of this study is to investigate the molecular mechanisms underlying neurodegenerative diseases.

### 2.1. Main Cause of Alzheimer’s and Parkinson’s Disease

Alzheimer’s disease and Parkinson’s disease are primarily caused by proteins, namely, β-amyloid in Alzheimer’s disease and α-synuclein in Parkinson’s disease [14]. Alzheimer’s disease is associated with various factors, including APP, PSEN1, PSEN2, and APOE. Gamma-secretase, which is encoded by PSEN1 and PSEN2, produces β-amyloid from amyloid precursor protein (APP), a membrane protein that helps in neuronal cell growth and repair. In the non-amyloid pathway, α-secretase and gamma-secretase cleave the amyloid precursor protein, but in the amyloid pathway, β-secretase and gamma-secretase cleave the amyloid precursor protein, producing β-amyloid. The accumulation of β-amyloid as multimers forms amyloid plaques, which disrupt signal transduction and cause neuronal cell death (Figure 1) [15]. Another factor is the tau protein, which helps microfiber proteins stay anchored inside nerve cells. When phosphorylated, tau protein aggregates and forms neurofibrillary tangles, disrupting normal nerve cell function and leading to apoptosis [16]. These changes lead to brain atrophy, affecting memory and learning [17]. The APOE gene is another major factor in Alzheimer’s disease. The APOE gene has three alleles, epsilon 2, epsilon 3, and epsilon 4, with varying risks of Alzheimer’s disease. Heterozygosity of APOE epsilon 4 increases the risk of dementia by 3–4 times compared to homozygosity of APOE epsilon 3, while homozygosity of APOE epsilon 4 increases the incidence of dementia by 12–15 times. APOE also binds with amyloid plaques, worsening the condition [18].

α-synuclein is the main causative agent of Parkinson’s disease. It is a protein encoded by the SNCA gene, and the monomer of this protein does not cause any significant problems. However, mutations in the SNCA gene can cause misfolding of α-synuclein monomers, which can lead to accumulation and aggregation, ultimately forming Lewy bodies. These Lewy bodies are neurotoxic to neurons, causing apoptosis or interfering with the transmission of signaling substances between neurons. Misfolded α-synuclein can also be produced by other factors such as mutations in mitochondrial-related proteins DJ-1, PINK1, and Parkin, which can cause mitochondrial dysfunction and generate reactive oxygen species (ROS). ROS can further produce misfolded α-synuclein or accumulate excessive toxicity in neurons, leading to cell death. Additionally, oxidation of dopamine produced and secreted by nerve cells can also cause α-synuclein aggregation, which can contribute to Parkinson’s disease (Figure 2) [19].

To eliminate toxic proteins, nerve cells typically rely on either autophagy or the ubiquitin proteasome system. If a protein is produced excessively or erroneously, it may be secreted out of the cell. In such cases, microglia or astrocytes perform phagocytosis or endocytosis to bring the secreted proteins back into the cells, where they are then removed through proteolysis. However, there is a limit to the cells’ ability to break down abnormal proteins or maintain normal protein levels, and when neuronal cells fail to maintain protein homeostasis, they eventually undergo apoptosis. This suggests that the brain gradually becomes ill over time [20,21,22].

### 2.2. Mitochondrial Dysfunction

Mitochondria are responsible for producing energy by using the electron transport chain to generate ATP through oxidative phosphorylation. However, one hallmark of neurodegenerative diseases is the dysfunction of mitochondria in neurons [11]. Mitochondria are vital for supplying ATP to cells through oxidative phosphorylation (OXPHOS), synthesizing biologically essential molecules, and catalyzing various redox reactions. Inefficient OXPHOS can result in mitochondrial dysfunction by producing reactive oxygen species (ROS). While low concentrations of ROS are crucial for normal cell signaling, prolonged or excessive exposure to ROS can lead to oxidative damage to macromolecules such as DNA, lipids, and proteins, ultimately causing cell death [23]. Moreover, it is believed that decreased oxidative phosphorylation can trigger apoptosis by inducing bioenergy depletion in neurons. Although a direct link between respiratory defects and neurons has yet to be established, mitochondrial respiratory deficits have been identified in several neurodegenerative diseases such as Alzheimer’s and Parkinson’s [24].

### 2.3. Inflammation

Neuroinflammation is an inflammatory response that specifically affects the brain and spinal cord. It can be triggered by factors such as disease, injury, infection, and stress, much like any other type of inflammation. Microglia and astrocytes are key regulators in neuroinflammation [25]. Recent studies have suggested a strong connection between neurodegenerative diseases and neuroinflammation. While acute inflammation of the brain functions as a defense mechanism against infection and injury, chronic inflammation like that seen in Alzheimer’s disease impairs the anti-inflammatory response. Chronic inflammation is caused by cytokines secreted by immune cells [26]. In patients with Alzheimer’s disease, sustained inflammatory responses are accompanied by the presence of β-amyloid plaques. This triggers the production of nitric oxide (NO) and reactive oxygen species (ROS) and promotes neuronal cell death by producing inflammatory cytokines, chemokines, and prostaglandins [27]. Microglia and astrocytes can detect β-amyloid by activating various sensor proteins such as TLR and NLR and then activate an inflammatory response through NF-kB. Recent research has also shown that β-amyloid can activate the NLRP3 protein, which is part of the NLR sensor of microglia, resulting in the production of IL-1b, an inflammatory cytokine [28].

A persistent immune response in the brain occurs not only in Alzheimer’s, but also in Parkinson’s. α-synuclein is a protein found at the synaptic terminal of nerve cells, but when α-synuclein aggregates to form Lewy bodies, it can activate immune cells. Microglia can phagocytose and degrade extracellular aggregated α-synuclein, and can produce inflammatory cytokines or ROS through NF-kB activation [29]. It has been reported that the immune function of microglia activated by α-synuclein is regulated by CD4+ T regulatory cells again [30].

Targeting the activation of immune cells by abnormal proteins such as β-amyloid or α-synuclein and the resulting neuroinflammatory response can be a potential strategy for treating neurodegenerative diseases such as Alzheimer’s or Parkinson’s [31,32].

## 3. Current Medications for Alzheimer’s and Parkinson’s Disease

As the population ages, the number of individuals with dementia worldwide is increasing, particularly among those over 60 years old. However, the multifactorial causes of Alzheimer’s and Parkinson’s disease are not fully understood [33,34]. In order to develop effective therapeutic agents for these neurodegenerative diseases, it is crucial to understand their molecular and biochemical pathogenesis. Several medications that modulate various molecular and biochemical mechanisms have been developed in the past [12]. Table 1 presents the medications that are currently used for the treatment of AD or PD.

### 3.1. Donepezil

Donepezil is a medication that works by inhibiting the enzyme acetylcholinesterase, which breaks down the neurotransmitter acetylcholine. By preventing the breakdown of acetylcholine, donepezil enhances cholinergic neurotransmission in the brain, thereby improving cognitive function in patients with Alzheimer’s disease [35]. Although donepezil does not halt the progression of the disease, it can improve symptoms such as memory loss, confusion, and problems with thinking and reasoning. Donepezil is approved by the FDA for use in mild-to-severe Alzheimer’s disease [36]. However, common adverse effects of donepezil include nausea, diarrhea, fatigue, dizziness, and insomnia.

### 3.2. Galantamine

Galantamine is an alkaloid that can be found in several plants, including daffodil bulbs, and is currently synthesized and provided as a medication. Similar to donepezil, galantamine functions as an acetylcholinesterase inhibitor and an allosteric modulator of nicotinic cholinergic neurotransmitter receptors [37]. Studies have demonstrated the efficacy of galantamine in treating cognitive symptoms in patients with mild-to-moderate Alzheimer’s disease. However, it is also associated with several major adverse effects such as convulsions, nausea, stomach cramps, vomiting, irregular breathing, confusion, and muscle weakness [38].

### 3.3. Rivastigmine

Rivastigmine is approved for the treatment of both Alzheimer’s and Parkinson’s disease [39]. Although the exact mechanism of action of rivastigmine is unclear, it is believed to work by inhibiting both acetylcholinesterase and butyryl cholinesterase. However, rivastigmine has major adverse effects such as abdominal pain, weight loss, poor gastrointestinal function, loss of appetite, and nausea. Overdosing can cause various symptoms such as irregular breathing, chest pain, and an irregular heartbeat [40].

### 3.4. Memantine

Memantine is a medication that functions as an antagonist of the N-methyl-D-aspartate (NMDA) receptor, and is prescribed to patients with moderate-to-severe Alzheimer’s disease. It works by preventing overactivation of glutamine receptors and slowing neurotoxicity by blocking NMDA receptors downstream of glutamate receptors. Some common adverse effects of memantine include pain, headache, fatigue, increased blood pressure, vomiting, drowsiness, coughing, and difficulty breathing [41].

### 3.5. Levodopa

Levodopa is a medication used to supplement dopamine deficiency in patients with Parkinson’s disease by acting as a precursor of dopamine. Once it passes through the blood–brain barrier, it is decarboxylated to dopamine by aromatic amino acid decarboxylase (AADC) and released from the presynaptic terminal of the striatum to compensate for the dopamine deficiency. However, dopamine has the potential to accelerate the disease by causing mitochondrial or lysosomal dysfunction and increasing α-synuclein oligomer concentrations, as suggested by some studies [42]. The long-term administration of levodopa has exercise-related adverse effects such as fluctuations, dyskinesias, and dystonia, and the non-motor adverse effects include autonomic dysfunction, mood control disorder, and cognitive decline [43].

### 3.6. Catechol O Methyltransferase Inhibitor (COMT)

COMT is an enzyme that catalyzes the methylation of catechol substrates using S-adenosyl-1-methionine as a co-factor. Various substances such as catechol, catecholamine, catecholestrogens, and ascorbic acid can serve as substrates for COMT [44]. The primary function of COMT is to eliminate catechols. In patients with Parkinson’s disease receiving levodopa/aromatic amino acid decarboxylase (AADC) inhibitors, levodopa is methylated to 3-O-methyldopa by COMT, which limits the availability of levodopa in the brain. COMT inhibitors prevent O-methylation of 3-O-methyldopa, a levodopa metabolite, and enhance the conversion of levodopa to dopamine in the brain [45]. However, COMT inhibitors also have adverse effects, such as movement disorders, confusion, hallucinations, urine discoloration, and diarrhea, due to the increased availability of levodopa [46].

### 3.7. Monoamine Oxidase-B Inhibitor

Monoamine oxidase-B (MAO-B) is a mitochondrial enzyme that is expressed widely in various tissues, including the stomach, liver, and nervous tissue. This enzyme is essential for the detoxification of amines by catalyzing the oxidative deamination of various monoamines and metabolizing the released neurotransmitters. The aldehydes produced during this process are metabolized by aldehyde dehydrogenase and aldehyde reductase to form glycols and carboxylic acids. However, the production of aldehydes with H_2_O_2_ suggests that the products of MAO action may be toxic to cells [47]. MAO inhibitors are medications currently being used to treat Parkinson’s disease by inhibiting the MAO enzyme. By improving the movement disorders caused by Parkinson’s disease, MAO inhibitors can help alleviate the symptoms of this condition. However, adverse effects such as nausea, dizziness, constipation, confusion, and hallucinations may occur with their use [48].

### 3.8. Dopamine Agonist

Dopamine agonists work by mimicking the effects of dopamine and directly activating dopamine receptors to improve the symptoms of Parkinson’s disease. There are two types of dopamine agonists: those derived from the fungus ergot and non-ergot agonists, both of which target dopamine D receptors [49]. Initially used as an add-on treatment for levodopa-induced dyskinesia [50], dopamine agonists are now also used to extend levodopa therapy and reduce the occurrence of motor complications [51]. However, ergot-derived dopamine agonists have adverse effects such as nausea, vomiting, orthostatic hypotension, hallucinations, and delusions, as well as adverse effects related to levodopa use [52].

**Table 1 ijms-24-08827-t001:** Medications for neurodegenerative diseases.

Target Disease	Medication Name	Mechanism	Adverse Effect	Ref.
Alzheimer’s disease	Donepezil	Acetylcholinesterase inhibitor	Nausea, vomiting, diarrhea, dizziness, trouble sleeping	[35,36]
	Galantamine	Acetylcholinesterase inhibitor and allosteric modulator on nicotinic acetylcholine receptors	Nausea, stomach cramps, vomiting, irregular breathing, confusion, muscle weakness	[37,38]
	Rivastigmine	Acetylcholinesterase inhibitor and butyrylcholinesterase inhibitor	Abdominal pain, weight loss, diarrhea, loss of appetite, nausea, irregular breathing, chest pain, irregular heartbeat	[39,40]
	Memantine	NMDA receptor agonist	Pain, headache, fatigue, increased blood pressure, vomiting, drowsiness, cough, shortness of breath	[41]
Parkinson’s disease	Levodopa	Supplement of dopamine	Fluctuations, dyskinesias, dystonias, autonomic dysfunction, mood control disorders, cognitive decline	[42,43]
	Catechol O methyltransferase inhibitor	Prolongation of levodopa action	Levodopa-related adverse effects, confusion, hallucinations, urine discoloration, diarrhea	[44,45,46]
	Monoamine oxidase-B inhibitor	Preventation of dopamine breakdown	Nausea, dizziness, constipation, confusion, hallucinations	[47,48]
	Dopamine agonist	Inducement of dopamine-like effects	Nausea, vomiting, orthostatic hypotension, hallucinations, delusions	[49,50,51,52]

## 4. Natural Materials—Compounds Derived from Natural Products

A search was conducted on the National Center for Biotechnology Information (NCBI) database to identify natural products or compounds derived from natural products that could be used as models for Alzheimer’s disease and Parkinson’s disease. The search keywords used were a combination of “Alzheimer’s disease” [MeSH Terms], “Parkinson’s disease” [MeSH Terms], and “Natural products” [MeSH Terms]. The search was not restricted by time, and papers that specified the mechanisms of action through both in vitro and in vivo studies were selected. The search results are presented in Figure 3 and Table 2.

The following documents were excluded based on the following criteria:A.Papers that did not contain the specified keywords.B.Review articles that covered multiple diseases.C.Case reports, clinical trial studies, and literature review studies.D.Abstracts and dissertations that were not relevant to the study.

### 4.1. Reynoutria multiflora Moldenke

*Reynoutria multiflora* Moldenke is a traditional medicinal herb in East Asia with antioxidant, anti-aging, and anti-inflammatory effects attributed to its various components. It also has a neuroprotective effect by reducing oxidative stress induced by H_2_O_2_ or glutamate. Studies have shown that extracts of *Reynoutria multiflora* Moldenke can prevent apoptosis by restoring FGF2 and BDNF expression in MPP+-induced neurotoxicity in SH-SY5Y cells. In mice models of Parkinson’s disease induced with MPTP, the oral administration of *Reynoutria multiflora* Moldenke extract activated the FGF2-Akt and BDNF-TrKB signaling pathways and restored dopaminergic neurons in the substantia nigra and corpus striatum [53].

### 4.2. Achillea fragrantissima Sch.Bip.

*Achillea fragrantissima* Sch.Bip. is a medicinal herb that has been traditionally used for the treatment of respiratory diseases, fever, obesity, and headaches. It has been found to inhibit the phosphorylation of stress-activated protein kinase/c-Jun N-terminal kinase (SAPK/JNK), extracellular signal-regulated kinase (ERK 1/2), and mitogen-activated protein kinase kinase (MEK1) as well as the transcription factor cyclic AMP response element-binding protein (CREB). This inhibition results in decreased accumulation of reactive oxygen species (ROS), reduced oxidative stress, and inhibition of cell death [54].

### 4.3. Theobroma cacao L.

*Theobroma cacao* L. a widely used raw material for chocolate, has been found to have antioxidant effects due to the presence of flavonoids such as flavan-3ols, epicatechin, and catechin. These flavonoids have been shown to inhibit free radical scavenging activity. Studies have demonstrated that cocoa extract can activate the BDNF/TrKB signaling pathway and exhibit cell protective effects in rat Pheocromocitoma PC12 cells treated with β-amyloid plaque or oligomers [55].

### 4.4. Salvia miltiorrhiza Bunge

*Salvia miltiorrhiza* Bunge has been reported to exhibit antioxidant and neuroprotective effects against neurotoxicity. In 6-hydroxydopamine-treated PC12 cells, it has been shown to increase Akt phosphorylation and activate the Nrf2 signaling pathway, leading to the upregulation of heme oxygenase-1 (HO-1), an antioxidant enzyme, thereby inhibiting ROS generation. In a zebrafish model of 6-hydroxydopamine-induced neurotoxicity, it demonstrated a neuroprotective effect by reducing neuronal cell death [56].

### 4.5. Asparagus racemosus Willd.

*Asparagus racemosus* Willd. is known for its antioxidant and immunomodulatory properties. In Swiss albino mice injected with kainic acid, it was found to inhibit excitotoxicity and reduce oxidative stress in hippocampal and striatal neurons. *Asparagus racemosus* Willd. also lowered lipid peroxidation and protein carbonyl content induced by kainic acid, and improved the activity of glutathione peroxidase (GPx) and glutathione (GSH) [57].

### 4.6. Opuntia ficus-indica (L.) Mill.

*Opuntia ficus-indica* (L.) Mill. are commonly found in the Mediterranean region and have been reported to exhibit antioxidant and anti-inflammatory properties. They promote the synthesis of heat shock proteins and prevent liver damage. Studies have shown that when administered to Saccharomyces cerevisiae with Aβ42(E22G) mutation, viability is improved. In addition, administration to Drosophila with human α-syn A53T mutation increases the survival rate. These plants also inhibit the fibrillogenesis of Aβ42 and α-syn proteins and prevent the disruption of lipid membrane integrity caused by the accumulation of oligomeric aggregates [58].

### 4.7. Gardenia jasminoides J.Ellis

*Gardenia jasminoides* J.Ellis, a plant commonly used in medicine and food in East Asia, has been found to have anti-inflammatory and antioxidant properties. In a study using the extract from this plant on APP/PS1 transgenic mice, it activated the phosphatidylinositide 3-kinase/AKT signaling pathway and demonstrated anti-neuroinflammatory effects by regulating the production of inflammatory proteins and cytokines [59].

### 4.8. Vitis labrusca L.

*Vitis labrusca* L. is commonly consumed as wine and has been shown to have beneficial effects on cardiovascular disease, cancer, and aging-related neurological diseases. In a Parkinson’s disease model using rats with 6-OHDA, the daily consumption of Vitis labrusca L. compound, procyanidin, led to reduced oxidative stress and improved mitochondria dysfunction, and PC12 cells treated with 6-OHDA showed neuroprotective effects via the activation of the PI3K/Akt signaling pathway [60].

### 4.9. Paullinia cupana Kunth

*Paullinia cupana* Kunth, also known as guarana, is a widely used traditional medicine and a common ingredient in energy drinks and foods due to its high caffeine content, a psychoactive pseudoalkaloid. It also contains polyphenols that provide various benefits. Studies have shown that it has antioxidant, antibacterial, and antigenotoxic effects. In Caenorhabditis elegans induced with AD, the administration of *Paullinia cupana* Kunth extract increases proteasome activity in muscles, prevents polyglutamine protein aggregation, inhibits cell death, prolongs cell lifespan, and delays paralysis. Additionally, it modulates antioxidant activity and proteostasis by reducing intracellular ROS and autophagosome accumulation, and increasing SOD-3 and HSP-16.2 expressions [61].

### 4.10. Tussilago farfara L.

*Tussilago farfara* L. has been found to exhibit anti-inflammatory properties. When PC12 cells are stimulated with H_2_O_2_ or 6-OHDA and treated with *Tussilago farfara* L. extracts, the Nrf2 pathway is activated, and the antioxidant HO-1 protects neurons from oxidative stress. In addition, in mice models with Parkinson’s disease induced by 6-OHDA injections, it improves motor function and prevents dopaminergic neuronal damage [62].

### 4.11. Panax ginseng C.A.Mey.

*Panax ginseng* C.A.Mey., an important medicinal material in East Asia, is known for its anti-inflammatory and antioxidant effects, as well as its potential to lower blood sugar and cholesterol and improve obesity. In PC12 cells treated with corticosterone, treatment with *Panax ginseng* C.A.Mey. extract increased cell viability and decreased apoptosis. Additionally, it reduced ROS generation and restored mitochondrial functions, including mitochondrial permeability transition pores and mitochondrial membrane potential. The expression of heat shock protein 90 and histone deacetylase 6, which are related to the glucocorticoid receptor, was increased, while the expression of endoplasmic reticulum stress-related proteins was decreased, thereby restoring endoplasmic reticulum function [63].

### 4.12. Polygala tenuifolia Willd.

*Polygala tenuifolia* Willd. is a medicinal herb traditionally used in East Asia for its potential to improve memory and cognitive function. It is known for its neuroprotective, neuroregenerative, antioxidant, and anti-aging effects. In SH-SY5Y cells treated with 6-OHDA, treatment with *Polygala tenuifolia* Willd. extract increased cell viability and decreased cell death. It protected the mitochondrial membrane potential and increased the expression of glutathione and superoxide dismutase, while reducing caspase-3 expression and increasing tyrosine hydroxylase expression to protect neurons [64].

### 4.13. Alpinia oxyphylla Miq.

*Alpinia oxyphylla* Miq. is a medicinal plant traditionally used to enhance memory and learning abilities affected by degenerative brain diseases. It has been found to protect neurons by reducing the expression of amyloid precursor protein, Aβ1-40, and Aβ1-42. Treatment with *Alpinia oxyphylla* Miq. extract activated Nrf2 through the Akt/GSK3b signaling pathway and inhibited oxidative stress in N2a/APP cells. In addition, administration of the extract to The Senescence Accelerated Mouse-Prone 8 mice showed the potential to delay cognitive function damage, enhance muscle strength, and restore motor ability [65].

### 4.14. Paeonia suffruticosa Andrews

*Paeonia suffruticosa* Andrews, a traditional medicine used for thousands of years in East Asia, has been reported to have anti-inflammatory, antioxidant, and anti-allergic effects. It inhibits inflammation, NO synthesis, and COX-2 expression in LPS-treated microglia. Furthermore, it inhibits ROS generation in oxidatively damaged cortical neurons treated with 6-OHDA, thereby reducing oxidative stress and increasing cell viability. This is achieved by increasing superoxide dismutase activity and anti-apoptotic protein expression [66].

### 4.15. Paeonia lactiflora Pall.

*Paeonia lactiflora* Pall., a traditional medicine used to treat cerebral ischemia, epilepsy, and degenerative brain disease, has shown potential neuroprotective effects. In glutamate-induced neurotoxicity in PC12 cells, treatment with *Paeonia lactiflora* Pall. extract increased cell viability by regulating the expression of Bcl-2 and Bax, which are proteins related to apoptosis. Furthermore, it also had a positive effect on mitochondrial function by regulating the mitochondrial membrane potential, which is often decreased in glutamate-induced neurotoxicity [67].

### 4.16. Cynanchum otophyllum C.K.Schneid.

*Cynanchum otophyllum* C.K.Schneid. has various uses, including medicine, cosmetics, and food, and has been traditionally used to treat inflammatory diseases such as epilepsy and rheumatism. Administering its extract to 3xTG Alzheimer’s model mice improved their learning and memory, reduced β-amyloid and tau aggregates, and inhibited microgliosis and astrocytosis. The extract showed a neuroprotective effect by activating the PPARα and TFEB signaling pathways, as well as activating the autophagy and lysosomal pathways [68].

## 5. Conclusions and Discussion

Neurodegenerative diseases are characterized by the death of nerve cells in the central nervous system, including the brain and spinal cord, and their incidence increases with age. Alzheimer’s disease and Parkinson’s disease are the two most common neurodegenerative diseases, but their exact causes and treatment methods are still being studied. In Alzheimer’s disease, β-amyloid has been a major focus of research, while in Parkinson’s disease, α-synuclein has been extensively studied. These pathogenic proteins disrupt the normal function of nerve cells, leading to inflammation, oxidative stress, and mitochondrial dysfunction. While efforts have been made to develop medications to treat these diseases, currently available medications only suppress symptoms and have many adverse effects, highlighting the need for new, more effective treatments. To investigate potential natural substances as the basis for new medications, a search was conducted using keywords related to Alzheimer’s disease, Parkinson’s disease, and natural products in the US National Center for Biotechnology Information. Sixteen papers were selected after screening, including ten related to antioxidants, four related to mitochondrial function, and others related to anti-inflammatory agents, self-destruction, and autophagy.

There are many differences between current drugs and natural products, as the research methods vary depending on when the drugs were developed or studied. The Alzheimer’s and Parkinson’s disease treatments listed in Table 1 are FDA-approved drugs, with donepezil approved in 1996, galantamine in 2001, rivastigmine in 2000, memantine in 2003, levodopa in 1975, tolcapone in 1998, entacapone in 1999, opicapone in 2020, pramipexole and ropinirole in 1997, and apomorphine hydrochloride in 2004. Most of these drugs were developed in the 1990s and 2000s and help ensure the proper transmission of neurotransmitters between nerve cells. However, current drugs have adverse effects and only delay the progression of symptoms, so there is a need for the development of new drugs [69,70,71,72,73,74,75,76,77,78,79].

Currently, there are ongoing clinical trials for drugs that aim to treat AD or PD with new mechanisms of action different from previously FDA-approved drugs. Examples of such drugs include mGluR5 silent allosteric modulator BMS-984923, c-Abl inhibitor IkT-148009, microbe lactobacillus plantarum PS128, LRRK2 inhibitor BIIB122, and sigma-2 receptor antagonist CT1812. Papers about these drugs can be found on NCBI, and clinicaltrial.org provides information about ongoing clinical trials. On the other hand, natural compounds are being studied with a different approach, aiming to activate neuronal cell recovery via Akt and BDNF signaling pathways, protect neuronal cells from ROS via Nrf2/HO-1 signaling pathways, or improve AD or PD via mitochondrial function recovery. However, natural compounds require more research to ensure their safety before undergoing clinical trials or being approved as drugs [80,81,82,83,84].

This study summarizes the neuroprotective effects of many natural products. While the keywords were restricted to Alzheimer’s disease and Parkinson’s disease, the mechanisms by which natural substances such as anti-inflammatory, antioxidant, and autophagy exhibit inhibitory effects are not specific to these diseases. This suggests that the efficacy of natural products that have been widely used to treat other diseases or to protect the body may also be applied to neurodegenerative diseases. Therefore, other natural products with similar efficacy also have potential applications. As natural products can be consumed as food in daily life, guidelines for daily intake should be established for more effective treatments.

## Figures and Tables

**Figure 1 ijms-24-08827-f001:**
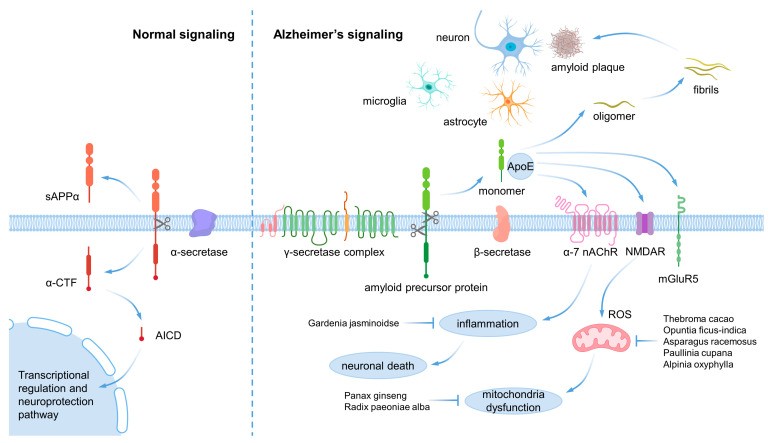
β-amyloid pathology in Alzheimer’s disease.

**Figure 2 ijms-24-08827-f002:**
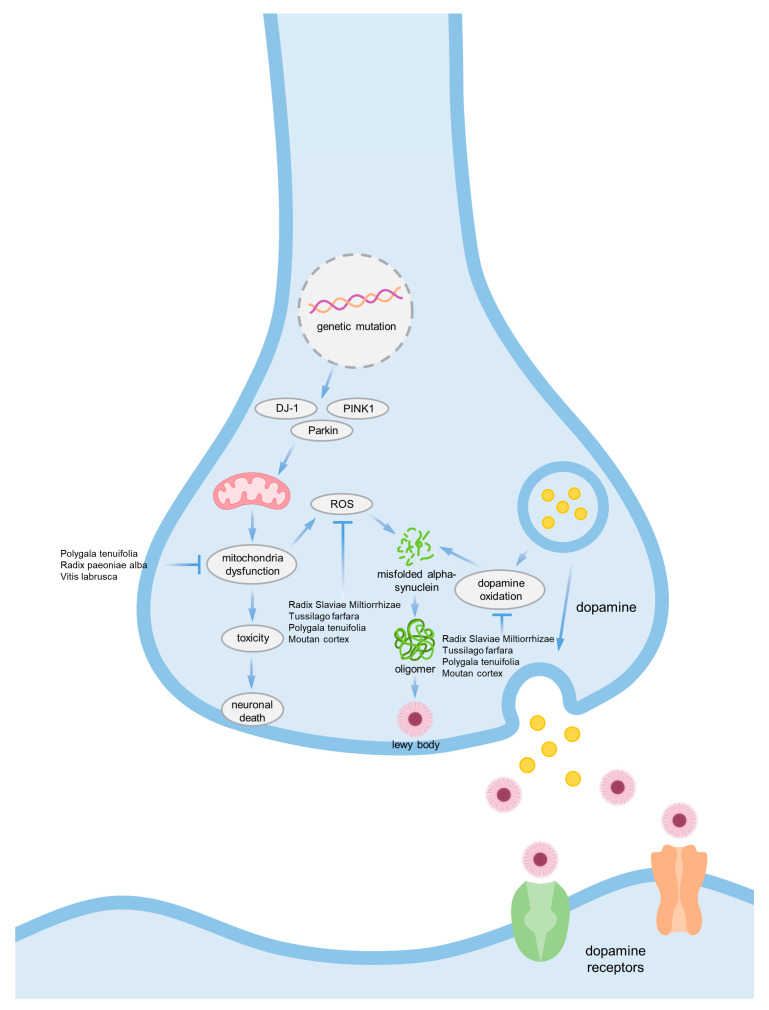
α-synuclein pathology in Parkinsons’s disease.

**Figure 3 ijms-24-08827-f003:**
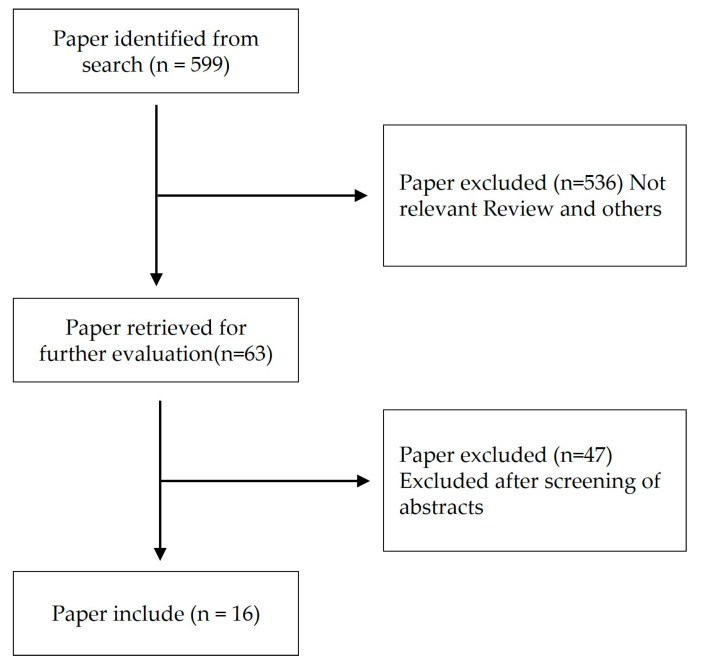
Flow chart of the study’s selection process.

**Table 2 ijms-24-08827-t002:** Effect of natural products for neurodegenerative diseases.

Natural Products	Part of the Natural Products	Model	Dose	Effects	Ref.
*Reynoutria multiflora* Moldenke	Purified compound	Mesencephalic dopamine neurons and SH-SY5Y cell	100, 200 μM (in vitro) 20 mg/kg (in vivo)	Anti-oxidative, anti-aging, and anti-inflammatory effects Restoration of the expression of FGF2 and BDNF, which inhibits apoptosis. Activation of the FGF2-Akt and BDNF-TrkB signaling pathways in the substantia nigra and corpus striatum, leading to the recovery of dopaminergic neurons.	[53]
*Achillea fragrantissima* Sch.Bip.	-	H_2_O_2_-treated astrocytes and neuron	Up to 34.7 μM (in vitro)	The inhibition of phosphorylation of stress-activated protein kinase/c-Jun N-terminal kinase (ERK 1/2), mitogen-activated protein kinase kinase (MEK1), kinase (SAPK/JNK), and the transcription factor cyclic AMP response element-binding protein (CREB) was observed.	[54]
*Theobroma cacao* L.	Commercial cocoa powder	β-amyloid-treated SH-SY5Y	Mixture of 30 μg/mL of epicatechin, 10 μg/mL catechin and 170 μg total polyphenols (in vitro)	Antioxidant, activating the BDNF survival pathway.	[55]
*Salvia miltiorrhiza* Bunge	-	6-OHDA-treated PC12 and zebrafish	100, 200, 400 μM	Activated the nuclear translocation of Nrf2 to increase heme oxygenase-1 (HO-1), conferring protection against ROS. Induced the phosphorylation of Akt.	[56]
*Asparagus racemosus* Willd.	Root	Intra-hippocampal and intra-striatal administration of kainic acid	18 mg/kg	Reduction of membranal lipid peroxidation and protein carbonyl following improvement in GPx activity and GSH contents.	[57]
*Opuntia ficus-indica* (L.) Mill.	Fruit skin	AD fly model with brain-specific expression of Aβ42 and PD fly model based on transgenic expression of the human α-syn A53T mutant	1 mg/mL (in yeast) 0.06% (in drosophila) 100, 400, 800, 2000 μg/mL (in vitro)	Inhibition of the fibrillogenesis of both Aβ42 and α-syn Accumulation of remodeled oligomeric aggregates that are less effective at disrupting lipid membrane integrity.	[58]
*Gardenia jasminoides* J.Ellis	Fruit	APP/PS1 transgenic mice	10, 20, 50 mg/kg	Suppressed neuroinflammatory responses in the brain through regulating phosphatidylinositide 3-kinase/AKT (PI3K/AKT) signaling pathway activation, expression of inflammatory proteins and release of inflammatory cytokines.	[59]
*Vitis labrusca* L.	Purified compound	6-OHDA-treated PC12 and rats	12.5, 25, 50 μM (in vitro) 60 mg/kg (in vivo)	Neuroprotection against 6-OHDA-induced neurotoxicity. Reduction oxidative stress and improvement in mitochondrial dysfunction. Activation of the PI3K/Akt signaling pathway.	[60]
*Paullinia cupana* Kunth	-	Aβ42-induced ad model of Caenorhabditis elegans	10, 50 mg/mL	Antioxidant activity and modulation of proteostasis. Intracellular ROS and the accumulation of autophagosomes reduction. Increased the expression of SOD-3 and HSP-16.2.	[61]
*Tussilago farfara* L.	Buds	6-OHDA-treated PC12 and mice	1.25, 2.5, 5, 10 μM (in vitro) 5 mg/kg (in vivo)	Activating the Nrf2/HO-1 signaling pathway.	[62]
*Panax ginseng* C.A.Mey.	Root	PC12 cells were treated with 250 μmol/L corticosterone	6.25, 12.5, 25, 50, 100, 200 μg/mL	Neuroprotection against corticosterone-induced damage in PC12 cells, and the intervening of HDAC6 and HSP90 of the GR-related function proteins, and subsequent restoration of ER and mitochondria functions.	[63]
*Polygala tenuifolia* Willd.	-	6-OHDA-treated SH-SY5Y	12.5, 25, 50, 100 μM	Antioxidative effects, maintenance of mitochondrial function, and regulation of caspase-3 and tyrosine hydroxylase expression and activity.	[64]
*Alpinia oxyphylla* Miq.	Purified compound	N2a/APP cells and SAMP8 mice	12, 25, 50, 100, 200, 400 μM (in vitro) 10, 20 mg/kg (in vivo)	Antioxidative effect through the Akt-GSK3b and Nrf2-Keap1-HO-1 pathways.	[65]
*Paeonia × suffruticosa* Andrews	Purified compound	6-OHDA-treated cortical neurons	0.75, 1, 1.5 μM	Decreased reactive oxygen species production. Increased cell viability, superoxide dismutase activity, and the anti-apoptotic protein expression.	[66]
*Paeonia lactiflora* Pall.	-	Glutamate-treated PC12 cell	0.1, 1, 10 μM	Neuroprotective effect on glutamate-induced apoptosis in PC12 cells by regulating the mitochondrial membrane potential and Bcl-2/Bax signal pathway.	[67]
*Cynanchum otophyllum* C.K.Schneid.		3XTg AD mice	6.5, 12.5, 25 μg/mL (in vitro) 25, 50, 100 mg/kg (in vivo)	Activation of PPARα-TFEB pathway.	[68]

## Data Availability

Not applicable.

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
