# Peer review of "Natural Products as the Potential to Improve Alzheimer’s and Parkinson’s Disease"

_ijms, 2023, doi:10.3390/ijms24108827_

Round 1

Reviewer 1 Report

Reviewer comment

The authors have conducted a broad and relatively comprehensive review of medications and natural product-induced effects on Alzheimer's disease and Parkinson's disease. The manuscript is well-written and effectively presents scientific information. However, there are a few areas that require correction to enhance the document's readability and ensure the accurate delivery of information.:

Section 3 requires the unification of terminology. The term "side effect" in the treatment drug description and "adverse effect" in Table 1 have different meanings, strictly speaking. It is necessary to harmonize the language and rectify any discrepancies. This also pertains to the language used in the abstract and discussion section.

Similar to the previous comments, the terms "drug" and "medication" are being used interchangeably. However, these two terms have distinct meanings, and it would be advisable to replace "drug" with "medication" in the main text for better clarity. It is necessary to provide a description of the abbreviations used throughout the text. Kindly include this information at the end of the document for reference.

Finally, authors should check the scientific names of plants by visiting the following sites.

http://www.worldfloraonline.org/

https://powo.science.kew.org/

Author Response

Listed are some comments regarding the submitted manuscript:.

  1. “Section 3 requires the unification of terminology. The term "side effect" in the treatment drug description and "adverse effect" in Table 1 have different meanings, strictly speaking. It is necessary to harmonize the language and rectify any discrepancies. This also pertains to the language used in the abstract and discussion section.” Thank you for feedback regarding the term mistake. As the reviewer's suggestion, we have standardized the term "side effect" to "adverse effect”.
  2. “Similar to the previous comments, the terms "drug" and "medication" are being used interchangeably. However, these two terms have distinct meanings, and it would be advisable to replace "drug" with "medication" in the main text for better clarity. It is necessary to provide a description of the abbreviations used throughout the text. Kindly include this information at the end of the document for reference.” Thank you for feedback regarding the term mistake. As the reviewer's suggestion, we have standardized the term "drug" to "medication”.
  3. Finally, authors should check the scientific names of plants by visiting the following sites. http://www.worldfloraonline.org/ https://powo.science.kew.org/ Thank you for feedback regarding the scientific name. We referred to the information on the official website suggested by the reviewer and reworded the scientific name accordingly.

Reviewer 2 Report

Dear author

Thank you for your valuable manuscript. Some comments mentioned for modification of your manuscript.

1. The scientific names must been write as trinomial based on theplantlist.org

2. How the scientific names were validated? Who was evaluated the traditional data with the morphology of the plants?

 3. How the meaning of two diseases have been evaluated in traditional medicine?

Best regards

Minor editing of English language required

Author Response

Reviewer #2:

  1. “The scientific names must been write as trinomial based on theplantlist.org”  Thank you for feedback regarding the scientific name. We referred to the scientific name by incorporating the information from the official website suggested by Reviewer 1 and reworded the scientific name accordingly.
  2. “How the scientific names were validated? Who was evaluated the traditional data with the morphology of the plants?” Thank you for the feedback on considerations of scientific verification that are important in natural products research. We selected relevant papers that aligned with our intended topic and organized the scientific names of the plants used as natural products based on the information provided in those papers. However, the original papers did not include information on the morphology of the natural products, which limited our ability to verify that aspect.
  3. “How the meaning of two diseases have been evaluated in traditional medicine?” We appreciate the authors' insightful feedback on the objectives of this study. It is well-known that Traditional Chinese Medicine (TCM) approaches the treatment of Alzheimer's disease and Parkinson's disease from a different perspective than Western medicine [Wang Z. et. al., 2013]. According to TCM, Alzheimer's disease is related to the aging of the five organs, qi, blood, yin, and yang. It is considered similar to deficiency syndrome, and treating it requires strengthening the spleen, promoting digestion, and improving blood circulation [Li S. et. al., 2021; Tan W. et. al., 2022]. In the case of Parkinson's disease, TCM views it as a condition in which the spleen and stomach lack energy. Therefore, treating Parkinson's disease with TCM involves using tonics to restore and strengthen the spleen and stomach, which can improve the patient's symptoms [Kum WF. et. al., 2011].

Reviewer 3 Report

I have the following comments for the authors:

1.     Were studies on plant-derived small molecules not included? Any reason for not including them? Same question for studies on purified polyphenols, flavonoids, alkanoids for eg. Curcumin etc

2.     In section 4.2, What part of the herb was tested in this study? Plant extract or a polyphenol or flavonoid?

3.     Table 2 should include a column specifying which part of the herb was used in the study i.e. leaf extract, root extract, fruit skin extract or rather a purified compound was used in the study?

4.     What dose of the plant extract was used in all the 16 references? Is there any commonality among the mechanism of action of these 16 herbs?

5.     Are any of these herbs available commercially as supplements?

6.     Page 14, line 468: This review article doesn’t “confirm” the neuroprotective effects of many natural products. It only reviews or highlight/summarize the neuroprotective effects of many natural products.

7.     In the discussion, talk about the differences and similarities between the mechanism of action of current drugs and natural drugs.

8.     Are there any combinatorial studies of these herbs? Or with structural modification of the plant-derived molecule or application of nanotechnology for decoration or delivery?

9.     What about the bioavailability of these plant extracts? Was this evaluated in any of these studies?

10.  Some of the references need to be updated, for eg. In section 4.8, for vitis labrusca, there is a latest study evaluating procyanidin (found in grapes) in Parkinson’s.  [Ying Zhang, Nanqu Huang, Mingji Chen, Hai Jin, Jing Nie, Jingshan Shi, Feng Jin, Procyanidin protects against 6-hydroxydopamine-induced dopaminergic neuron damage via the regulation of the PI3K/Akt signalling pathway, Biomedicine & Pharmacotherapy, Volume 114, 2019].

Author Response

  1. “Were studies on plant-derived small molecules not included? Any reason for not including them? Same question for studies on purified polyphenols, flavonoids, alkanoids for eg. Curcumin etc” I appreciate the important questions from the reviewer regarding the key factors determining the efficacy of plants. The active ingredients that demonstrate actual efficacy in natural products are likely to be small molecules such as polyphenols and flavonoids. However, this paper aims to provide a summary of natural products. In TCM, the plant itself or extracted components are used, rather than selectively using small molecules derived from plants, so this paper does not include information on small molecules.
  2. “In section 4.2, What part of the herb was tested in this study? Plant extract or a polyphenol or flavonoid?” Thank you for the feedback on considerations of scientific verification that are important in natural products research. This paper only provides information that "The wild sun-dried plant Achillea fragrantissima (37 gr) was homogenized and extracted twice with ethyl acetate and once with ethyl acetate:methanol (9:1).”. Unfortunately, the original papers did not include information on the morphology of the natural products, which limited our ability to verify that aspect.
  3. “Table 2 should include a column specifying which part of the herb was used in the study i.e. leaf extract, root extract, fruit skin extract or rather a purified compound was used in the study?” Thank you for your valuable feedback on our research on natural materials. there were only a few of the selected papers provided information on the specific plant parts used. In response to this, we have summarized the available information in Table 2 for easy reference.
  4. “What dose of the plant extract was used in all the 16 references? Is there any commonality among the mechanism of action of these 16 herbs?” We have updated Table 2 to include information on the doses of plant extracts used in the experiments. While it is important to note that not all plant extracts have the same mechanism of action, we have identified common mechanisms in some of the plants studied. One of the most frequently mentioned mechanisms is the Nrf2 signaling pathway, which plays a key role in protecting nerve cells. Another commonly identified mechanism is the Akt signaling pathway, which also has a protective effect on nerve cells.
  5. “Are any of these herbs available commercially as supplements?” Thank you for your feedback on follow-up considerations in R&D. Ginseng is already available in various forms as a supplement in many countries, and there are currently no products that combine ginseng with other natural products and have undergone bacterialization. The aim of this study was to demonstrate the efficacy of various natural substances in the development of treatments for Alzheimer's and Parkinson's diseases. We believe that this research has the potential to facilitate the discovery and development of therapeutic materials for these diseases. In the microbiological and ecological stages of development and production, the potential risks of bacterialization have been studied and many experts in the field are exploring ways to address these concerns.
  6. “Page 14, line 468: This review article doesn’t “confirm” the neuroprotective effects of many natural products. It only reviews or highlight/summarize the neuroprotective effects of many natural products.” We appreciate the important point made about presenting research content, and we agree with this opinion. Our intention was to demonstrate the experimental efficacy at the research level rather than to determine it. We have accordingly revised the relevant part of the content to use appropriate language.
  7. “In the discussion, talk about the differences and similarities between the mechanism of action of current drugs and natural drugs.” Thank you for your valuable comments, which can enhance the quality of this study. We have incorporated the suggestions provided by the reviewer into the discussion section.

“Are there any combinatorial studies of these herbs? Or with structural modification of the plant-derived molecule or application of nanotechnology for decoration or delivery?” We appreciate your feedback regarding the advanced follow-up development technology in the application of natural medicines. In response to the reviewer's recommendation, we have conducted additional research on studies that focus on augmenting the effectiveness of plant-derived molecules using modification or nanotechnology, as it pertains to the natural materials discussed in our paper. Our findings indicate that several studies have explored the enhancement of molecule efficacy through modification or nanotechnology in specific natural products. The outcomes of this investigation and the efficiency of the various methods are outlined in the table provided below.

Natural products

Modification of the plant derived molecule

Effects

Reference

Theobroma cacao

the partial deacetylation and de-esterification of pectin

Increase of the pro-inflammatory cytokine level (TNFa, IL-10, IL-12)

Juliana CA et. al., 2016

Salvia miltiorrhiza

New imidazole tanshinones

Vascular protective and restorative activity

The pro-angiogenesis effect

Zhe-Rui Z et. al., 2014

Natural products

Nanotechnology application

Effects

Reference

Salvia miltiorrhiza

Powder prepared using nanotechnology

Antioxidant bioactivity

Je RL et. al., 2008

Opuntia ficus-indica

HAP nanoparticle

Antimicrobial activity

Gopi D et. al., 2015

Paullinia cupana

Liposome

Antioxidant activity

Isabel R et. al., 2019

Panax ginseng

Gold and silver nanoparticles

In gold nano particle, Antioxidant activity, highly biocompatibility, and anti-inflammatory effect

In silver nanoparticle, antioxidant activity, anticancer effect, and anti-inflammatory effects

Priyanka S et. al., 2016

  1. “What about the bioavailability of these plant extracts? Was this evaluated in any of these studies?” Thank you for your valuable comments on practical development in the study of natural product formulations. While the relevant papers we selected did not include bioavailability studies, other studies have confirmed the bioavailability of certain natural products. For instance, Xin Z et al. (2020) reported bioavailability percentages of 3.25%, 2.95%, 2.36%, 1.17%, and 42.91% for Sibiricose A5, sibiricose A6, 3,6’-disinapoyl sucrose (DSS), tenuifoliside A (TFSA), and 3,4,5-trimethoxycinnamic acid (TMCA), respectively, in their study on Polygala tenuifolia. Cristina MD et al. (2020) found the bioavailability of Paullinia cupana to be approximately 40%. Furthermore, Sharleen C et al. (2013) reported bioavailability percentages of 4.23%, 32.32%, and 27.17%, respectively, for geniposide, Gardenia jasminoides fruit extract, and Gardenia herbal formulation in their study on Gardenia jasminoides, which was administered orally to rats.
  2. “Some of the references need to be updated, for eg. In section 4.8, for vitis labrusca, there is a latest study evaluating procyanidin (found in grapes) in Parkinson’s. [Ying Zhang, Nanqu Huang, Mingji Chen, Hai Jin, Jing Nie, Jingshan Shi, Feng Jin, Procyanidin protects against 6-hydroxydopamine-induced dopaminergic neuron damage via the regulation of the PI3K/Akt signalling pathway, Biomedicine & Pharmacotherapy, Volume 114, 2019].” Thank you for your valuable comments, which can enhance the reliability of this study. We have incorporated your suggested content and revised the existing content accordingly to improve the overall quality of the study.
